# Perceived Application and Barriers for Gait Assessment in Physical Therapy Practice in Saudi Arabia

**DOI:** 10.3390/life13010050

**Published:** 2022-12-24

**Authors:** Salhah Hobani, Anas Mohammed Alhakami, Shadab Uddin, Fuzail Ahmad, Hana Alsobayel

**Affiliations:** 1Health Rehabilitation Sciences Department, College of Applied Medical Sciences, King Saud University, Riyadh 11437, Saudi Arabia; 2King Faisal Medical City for Southern Regions, Medical Rehabilitation Center, Abha 62527, Saudi Arabia; 3Department of Physical Therapy, Faculty of Applied Medical Sciences, Jazan University, Jazan 45142, Saudi Arabia; 4Faculty of Applied Sciences, Al Maarefa University, Ad Diriyah, Riyadh 13713, Saudi Arabia

**Keywords:** gait assessment, gait analysis, barriers, physical therapist, clinical practice, assessment tools, musculoskeletal, Saudi Arabia

## Abstract

Introduction: Gait is a major function of independence that determines the quality of life, participation, and restriction. Gait assessment in physical therapy practice is fundamental for assessing musculoskeletal conditions. This study planned to determine the extent and barriers of using gait assessment tools by physical therapists in clinical practice in Saudi Arabia. Method: A cross-sectional design was used. A standardized survey was sent through e-mail and social media to physical therapists working in hospitals, clinics, and rehabilitation centers in different Saudi Arabian regions. Results: A total of 320 physical therapists from different regions in Saudi Arabia participated. There was a significant relationship between using gait assessment tools by physical therapists and patient groups (*p* = 0.002), receiving training (*p* = 0.001), dealing with patients who suffer from gait problems (*p* = 0.001), and visiting the gait laboratory (*p* = 0.001). Physical therapists’ use of gait assessment tools for musculoskeletal conditions in clinical practice in Saudi Arabia was limited. The primary barrier preventing physical therapists from using gait assessment methods in clinical practice was a lack of resources, including instruments, space, time, and funds.

## 1. Introduction

Gait, a mode of locomotion in which the two legs alternately offer both support and propulsion [1], is a crucial component of independence that affects one’s quality of life, level of engagement, and degree of constraint. Gait depends on the functional integrity of the musculoskeletal, nervous, and cardiorespiratory systems [2]. Human gait changes because of normal aging changes and medical, psychiatric, neurological, and musculoskeletal conditions [2,3].

Musculoskeletal conditions among adults are one of the most common factors that impact the individual’s gait, function, and health state [4,5,6]. Around 1.7 billion people in the world have musculoskeletal conditions that are considered the leading causes of mobility limitations and disability [7]. Gait assessment is a well-established measuring tool to assess gait, functional activities, and the state of injuries overtime [8,9,10]. This includes instrumental gait analysis tools employing Electromyography (EMG), force plates, 3-Dimensional gait analysis systems, video recording, and Observational Gait Assessment tools employing video recording or scales such as the Chamorro Assisted Gait Scale for musculoskeletal injuries, the Standardized Gait Score for tibia fracture and musculoskeletal disorders, etc. [11,12]. 

Walking speed, stride and step length, swing and stance durations, and other force-related metrics such as GRF, muscle force, and joint momentum are examples of parameters that offer a complete description of gait mechanics. Gait analysis has a significant influence in a variety of domains, including human identification, sports, and, most notably, clinical medicine, where objective gait analysis plays a vital part in the diagnosis, treatment, and monitoring of neurological, cardiopathic, and age-related illnesses [11,12].

Gait assessment is essential in rehabilitation and physical therapy practice for assessing gait parameters, and the body of data indicates that it has a high degree of technical accuracy and diagnostic potential, which influences treatment decision-making based on the consistency of the gait analysis and clinical data [8,9,10,13]. Previous studies have shown a lack of awareness regarding gait assessment among physical therapists, and many do not use any tools or methods of gait assessment [14,15]. Physical therapists frequently used gait assessment tools for pediatric and neurologic conditions [14,15,16], with less focus on musculoskeletal conditions [17,18].

Gait analysis examines an individual’s ambulatory pattern and allows for the measurement of the location and orientation of distinct body segments in space, as well as insight into the forces responsible for the observed segmental displacements. Many gait assessment tools are applicable in routine clinical practice for musculoskeletal conditions. It is unknown whether physical therapists use gait assessment tools for musculoskeletal conditions in clinics in Saudi Arabia. There is a lack of studies examining the extent and barriers of using gait assessment tools among physical therapists in Saudi Arabia. To the best of our knowledge, the extent to which physical therapists use gait assessment tools in Saudi Arabia has still not been addressed. Therefore, this study aimed to determine the extent and barriers of using gait assessment tools for musculoskeletal conditions by physical therapists in clinical practice in Saudi Arabia.

## 2. Methods

### Participants and Design

A cross-sectional study was conducted. King Saud University’s Institutional Review Board (IRB) Ethical Committee granted the study approval (NO. E-205486). The inclusion criteria involved (1) licensed physical therapists working in outpatient physical therapy clinics in either private or governmental sectors in Saudi Arabia and (2) managing adult cases with musculoskeletal problems. We excluded undergraduate students, interns, and academic or administrative physical therapists who do not deal with patients (non-clinicians). The representative sample needed for this study was estimated to be 362 physical therapists. The estimate was made based on a response rate of 50%, a level of confidence of 95%, and a margin of error of 5%. The Saudi Commission for Health Specialties states that there are 6028 physical therapists in Saudi Arabia [19].

The survey used in this study was based on the developed and validated questionnaire by Toro et al. [15]. The survey is examined:(1)Demographics of physical therapists.(2)Physical therapist experience and training.(3)Use of gait assessment tools in practice.(4)Physical therapist perspectives on gait assessment tools.

It consisted of five sections; the first and two sections dealt with the demographic data: nationality, gender, age; academic qualification, years of experience, geographical regions; using gait assessment tools; patient groups; and dealing with gait problems. Parts 3–5 addressed the experience of gait assessment, gait assessment tools used, and physical therapist opinions and views on the gait assessment tools. Questions were presented as yes-or-no questions, multiple-choice questions, and ordinal rating scales. A few changes have been made that are not applicable anymore, such as the standard VHS video camera, television, and floppy disk. The English language version of the questionnaire is the only one used since the English language is predominantly used in most of the curriculum courses of physical therapy schools in Saudi Arabia. Permission was obtained from the original author to use the same questionnaire in this study.

Participants were recruited through e-mail and web-based media (Twitter and WhatsApp) and sent to outpatient physical therapy clinics in different regions of Saudi Arabia. The survey was also sent online directly to the potential participants. The survey was administered from January 2021 to February 2021. The online survey is cost-effective, saves time, is easy to cover many geographical regions, and enhances the ability to reach therapists located away [20]. This option also allows participants to answer in their own time [20]. To ensure an increased response rate, the Saudi Physical Therapy Association (SPTA), a local professional society with branches in most regions of Saudi Arabia, was approached to help distribute the questionnaire among their databases. Reminders were sent three days and two weeks later to ensure a response. A Google Form was set up and used to collect data. The data was collected only by the primary investigator, who has a secure Google account and e-mail connected directly with an electronic survey link to ensure data confidentiality. Before beginning the survey, participants were asked to indicate whether they would participate in the study by checking one of the boxes on the first page of the consent form.

The data from the Google Form related to therapists’ responses was saved on Microsoft Excel, revised, coded, and then entered into the statistical software IBM SPSS version 22 (Inc. Chicago, IL). All statistical analysis was carried out using two-tailed significance tests. A *p*-value less than or equal to 0.05 was considered to be a statistically significant level. Cross-tabulation was used to test the distribution of the gait assessment tool by demographic and experience with gait assessment data. A Pearson chi-square test was used to examine the relationship between the gait assessment tool and physical therapists’ demographics and experience with gait assessment data. Logistical regression was used to predict the likelihood of using gait assessment tools in the management of musculoskeletal patients with gender, educational background, years of clinical experience, patient type, perceived need for gait assessment tools, and gait assessment training as predictor.

## 3. Results

A total of 320 physical therapists responded and completed the survey out of 362, with a response rate of 88.3%. Table 1 shows the demographic data and response to the survey question by the participants. Most respondents were females, 68% (n = 218), and held bachelor’s degrees, 78.4% (n = 251). A high percentage of the participants, 38.4% (n = 123), reported practicing physical therapy for 3–5 years. Nearly 52% (n = 166) of physical therapists did not use gait assessment tools in their clinical work, even though 88% (n = 283) of their patients had gait problems. Only 38.2% (n = 108) of physical therapists received formal training in gait assessment out of all those who dealt with patients who suffered from gait problems.

### Gait Assessment Tools and Barriers 

Most physical therapists 42% (n = 120) used visual observation tools. At the same time, around 60% (n = 168) of physical therapists reported that they did not have gait assessment tools in their department. The most common barrier reported by respondents, at 53.5% (n = 151), was having no tools available, followed by lack of space at 28.7% (n = 81) and time at 22.3% (n = 63). Table 2 outlines the gait assessment tools and barriers to use gait assessment tools.

Physical therapists’ likelihood of employing gait assessment tools in the management of musculoskeletal patients was analyzed by means of logistical regression, with variables including gender, educational background, years of clinical experience, patient type, perceived need for gait assessment tools, and gait assessment training. Statistically, the logistic regression model was significant; χ2 (6) = 18.308, *p* = 0.006. The model successfully identified 60.4% of instances and explained 84.0% (Nagelkerke R2) of the variation in the usage of gait assessment tools. 

According to Table 3, we can conclude that the likelihood that a physical therapist will choose a gait assessment tool is 0.47 times greater among those who treat all type of patients than among those who treat just musculoskeletal patients. Similarly, between physical therapists with and without training in gait assessment tools, those with training were 0.48 times more likely to use the gait assessment tools. Gender, qualification, clinical experience, and perceived need for gait assessment tools were not found to be influencing the decision to use gait assessment tools.

## 4. Discussion

The current study aimed to determine the extent and barriers of using gait assessment tools for musculoskeletal conditions by physical therapists in clinical practice in Saudi Arabia. This problem might be the result of poor gait assessment technique for patients with gait problems in clinical practice, which could have an impact on patient treatment and results over time. A lack of availability of tools was the main barrier reported by physical therapists, which could be another possible explanation for the low rate of using gait assessment tools. The current study’s results were in line with those of studies carried out in other countries, including the UK [15] and Ireland [14], which also produced comparable results.

It was noted that almost all physical therapists with a higher qualification used gait assessment tools more frequently than others, which might be a result of their more specialized areas of practice, knowledge, awareness, and skills [14,15]. When dealing with heterogeneous patient groups, a substantial majority of physical therapists did not employ any gait evaluation instruments. This might be due to a lack of expertise and awareness regarding the employment of gait evaluation methods for disorders other than musculoskeletal problems in Saudi physical therapy clinics. The findings were similar to physical therapists who did not use any gait assessment tools in the UK and Ireland, despite dealing with patients who suffer from gait problems [15,16]. It has been said that using gait assessment tools in clinical practice depends on increasing knowledge and awareness of therapists [21,22].

The survey found that a comparatively lower number of physical therapists in Saudi Arabia had received formal gait assessment training, which may have hampered their ability to use these tools efficiently in clinical practice, suggesting the need of gait assessment education at all levels of study, but particularly at the undergraduate level. Our findings were similar to those of British physical therapists [15] and in contrast to those of physical therapists in Ireland who had received formal training [14].

Few physical therapists used standardized gait assessment tools in the current study and in previous studies in the UK and Ireland [14,15]. Al-Muqiren et al. [23] reported that 62% of physical therapists in Saudi Arabia used standardized outcome measures (SOMs) with different conditions in physical therapy practice, such as 6 MWT, 10 MWT, and others. These measures have different purposes in assessing gait and functions. The rate of using specific SOMs (e.g., gait speed and functional ability tests) was also low among Canadian physical therapists in clinical practice [22]. This rate was due to the lack of knowledge and awareness, which were the common barriers to using assessment tools for neurological conditions among Canadian therapists [22]. However, Harradine et al. [24] reported that there is still a lack of standardized observational gait assessment tools or systems for musculoskeletal conditions in adults in clinical practice. Wearable systems and instrumental gait analysis demonstrated greater potential for assessing gait parameters and functions of musculoskeletal conditions such as spinal stenosis or knee OA than outcome measures of function such as the Oswestry Disability Index (ODI) and Western Ontario & McMaster Universities Osteoarthritis Index (WOMAC) [8,9,25]. These measures could reduce subjectivity or bias in reporting [8,9].

In the current study, approximately 60% of physical therapists did not have gait assessment tools in their department, which was the most common barrier in clinical practice. This could be because physical therapists’ knowledge of gait assessment is limited, and gait assessment tools, particularly instrumental gait analysis, are typically found in academic settings rather than clinical settings [15,26]. As reported by physical therapists in the current study, lack of space, time, and budget constraints were common barriers. Similarly, the most frequent barriers were stated by British and Irish physical therapists [14,15]. In a systematic review, Jang et al. [21] reported similar barriers among therapists who used balance and gait assessment tools for neurological patients in clinical practice. They categorized them mainly into individual, environmental, and measure-specific barriers. As a result of these problems, it might be hard for physical therapists in Saudi Arabia to choose the right assessment tools and treatments for their patients.

The confidence rate of the majority of physical therapists to assess gait visually was moderate (rating 3 out of 5), which was similar to previous studies [14,15]. It may mean that they still do not have enough skills to assess gait. These issues have been significantly related to receiving training and higher qualifications for extended study [14,15]. Gaining knowledge from evidence about the structures of gait assessment tools and implementing them in practice, years of working with patients with gait problems, and clinical resources may make physical therapists familiar with using appropriate gait assessment tools. As a result, their confidence may be increased.

The results also showed that most physical therapists need guidelines, protocols, and newly developed tools to use gait assessment tools in clinical practice, which is in agreement with the findings in Ireland and the UK [14,15]. This may reflect that training may increase awareness and implementation of gait assessment in clinical practice when patients are contacted directly. The use of gait assessment tools is also related to training and visiting the gait laboratory. In a systemic review, Stander et al. [27] reported that training programs for transferring knowledge from evidence to practice are more effective for physical therapists in providing clinical guidance and improving the quality of care.

In our study, all physical therapists who used gait assessment tools reported that the acquired gait-related data provided a better understanding of clinically meaningful outcomes such as the prediction and discrimination of neurological disorders from healthy controls, monitoring disease progression, identifying fall risk and freezing of gait, and assessing treatment and rehabilitation efficacy. As gait analysis technologies provide immediate feedback, the process of building a customized therapy is simplified, allowing physical therapists to assist their patients prevent or reduce the severity of future injuries by evaluating data on the body’s response to motions and the limbs’ reaction to force [21,22].

### 4.1. Study Limitations

This study has some limitations. The study design was cross-sectional based on the web-based media and online survey to collect data, which may not fully reflect the real answers of participants. The study sample was a convenience that may not represent the whole physical therapist population in Saudi Arabia. It also included physical therapists who mostly dealt with musculoskeletal conditions, which would restrict the applicability of the findings to Saudi Arabian practitioners of other disorders, while most physical therapists do not solely deal with musculoskeletal conditions. We did not examine the validity of the questionnaire, which may require updating to include the new technologies.

### 4.2. Recommendations and Future Direction

Clinical associations and facilities of physical therapy in Saudi Arabia should make an effort to provide strategies to promote and raise awareness of using gait assessment tools by enhancing university curricula and educational training programs either at undergraduate or postgraduate levels. This essential effort can improve clinical assessment and treatment to improve patients’ quality of healthcare. Physical therapists should consider gait assessment tools when evaluating and monitoring musculoskeletal patient outcomes.

Future studies must examine the psychometric properties of newly developed gait assessment tools with different musculoskeletal conditions. In addition, guidelines or protocols for using gait assessment tools in clinical practice should be provided. Moreover, future studies should consider how continuing education and training programs affect physical therapists’ ability to use gait assessment tools, make changes to clinical decisions, and choose the right treatments.

## 5. Conclusions

Gait assessment tools are important for providing valuable information in physical therapy practice. The extent of the use of gait assessment tools for musculoskeletal conditions by physical therapists in clinical practice in Saudi Arabia was limited. However, the majority of them dealt with patients who had gait problems. It appears that visual observation was the most common tool used. Patient groups, training, dealing with patients with gait problems, and visiting the gait laboratory were the key factors related to using gait assessment tools. A lack of availability of tools, space, time, and budgetary constraints were the main barriers that constrained physical therapists from using gait assessment tools in clinical practice. There is a strong demand to provide guidelines, protocols, and training programs at the clinical site of gait assessment. Physical therapists require newly developed gait assessment tools that can be used quickly and easily. Most physical therapists were willing to use gait assessment tools in the future.

## Figures and Tables

**Table 1 life-13-00050-t001:** Demographic and work related data of participants.

Demographics	(n = 320)	%
Region		
Eastern	18	5.6%
Middle	85	26.6%
Northern	25	7.8%
Southern	131	40.9%
Western	61	19.1%
Age in years		
21–30	215	67.2%
31–40	84	26.3%
41–50	17	5.3%
>50	4	1.3%
Gender		
Male	102	31.9%
Female	218	68.1%
Nationality		
Saudi	294	91.9%
Non-Saudi	26	8.1%
Qualification		
Diploma	5	1.6%
Bachelor	251	78.4%
Master	54	16.9%
PhD	10	3.1%
Years of clinical experience you have as a physical therapist		
0–2	65	20.3%
3–5	123	38.4%
6–10	89	27.8%
>10	43	13.4%
Use gait assessment tools in clinical practice		
Yes	154	48.1%
No	166	51.9%
The patient group/s you work with		
Only Musculoskeletal patients	69	21.60%
Mixed patients (different conditions)	251	78.40%
Deal with patients who suffer from any gait problems		
Yes	283	88.40%
No	37	11.60%
Time duration (years) you have dealt with gait disorders (n = 283)		
0–2	132	46.60%
3–5	73	25.80%
6–10	36	12.70%
>10	42	14.80%
Received any formal training in gait assessment (n = 283)		
Yes	108	38.2%
No	175	61.8%
Any of your patients visited a gait laboratory for a full gait assessment (n = 283)		
Yes	40	14.1%
No	156	55.1%
Don’t know	87	30.7%
You were able to understand the findings (n = 40)		
Yes	35	87.5%
No	5	12.5%
The findings influenced your clinical practice (n = 40)		
Yes	34	85.0%
No	6	15.0%

**Table 2 life-13-00050-t002:** Gait assessment tools and barriers.

Variables		(n = 283)	%
Are you using any of these gait assessment tools/methods	A standardized gait assessment form	80	28.3%
	An ‘in-house’ developed gait assessment tool	31	11.0%
	Visual observation form/chart	120	42.4%
	Still Photography	15	5.3%
	Video Record (Digital video camera)	36	12.7%
	None of the above	89	31.4%
What are your barriers for using a gait assessment tool?	There is no tool available for my patient group	151	53.5%
	Lack of interest	23	8.2%
	Budget constraints	32	11.3%
	Lack of space	81	28.7%
	I am not aware of any gait assessment tool	26	9.2%
	Lack of time	63	22.3%
	I am working in unsuitable environment	32	11.3%

**Table 3 life-13-00050-t003:** Logistical regression predicting the likelihood of employing gait assessment tools in the management of musculoskeletal patients by Physical therapists based on their gender, qualification, clinical experience, type of patients, perceived need for gait assessment tools and training for gait assessment.

Predictors	B	Wald χ2	*p*	EXP (B)	95% C.I. for EXP (B)	
					Lower	Upper
Gender	−0.227	0.710	0.399	0.797	0.469	1.352
Qualification	−0.325	1.045	0.307	0.722	0.387	1.348
Clinical Experience	0.034	0.017	0.895	1.034	0.628	1.703
Type of Patient	−0.752	5.354	0.021	0.471	0.249	0.891
Perceived need for gait assessment tools	0.357	0.896	0.344	1.429	0.682	2.993
Training for Gait Assessment	−0.738	7.919	0.005	0.478	0.286	0.799

B: Regression constant; Wald χ2: Wald Chi Square constant; P: Significance level; EXP (B): Odds Ratio; C.I.: Confidence Interval.

## Data Availability

The data presented in this study are available on request from the corresponding author.

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
