# Peer review of "Perceived Application and Barriers for Gait Assessment in Physical Therapy Practice in Saudi Arabia"

_life, 2022, doi:10.3390/life13010050_

Round 1

Reviewer 1 Report

I appreciate the opportunity to review this manuscript on the „Perceived Application and Barriers in Saudi Arabia for Gait Assessment in Physical Therapy Practice “.

 A cross-sectional study was conducted by the authors. Participants were recruited through e-mail and web-based media. The results showed that the extent of using gait assessment tools by physical therapists in clinical practice was less than 50 % in Saudi Arabia.

 The authors state that gait assessment tools are important to provide valuable information in physical therapy practice. The extent of the use of gait assessment tools for musculoskeletal conditions by physical therapists in clinical practice in Saudi Arabia is limited

 There are some shortcomings which have to be addressed before taking this manuscript into consideration for publication:

·       What is the goal of the testing? What do the authors mean with gait assessment tools? Please clarify

·       What kind of activities are involved?

      Please clarify reasons why you may need a physical therapist to do a balance and gait assessment

·       How do you assess gait analysis?

·       What is included in a gait assessment?

·       How is a normal gait described?

·       Which parameters are measured?

Overall: This study does not really contribute valuable information to the preexisting studies and literature.

Author Response

Response to 1st Reviewer comments

  1. What is the goal of the testing?

Authors Response: Gait assessment is essential in rehabilitation and physical therapy practice for assessing gait parameters, and the body of data indicates that it has a high degree of technical accuracy and diagnostic potential, which influences treatment decision-making based on the consistency of the gait analysis and clinical data. (Page 3, line 108-111)

  1. What do the authors mean with gait assessment tools? Please clarify

Authors Response: The introduction has been revised to include the gait analysis tools. (Page 3, line 94-99)

  1. What kind of activities are involved?

Authors Response: Activities include functions as walking, lifting, standing, climbing stairs, running, double or single limbs squat, rotation, jumping, motion of body joints (flexion, extension, abduction, adduction, tilting). But in this study, we don’t measure gait parameter and activities. We just explore the state of using gait assessment tools among physical therapists.

  1. Please clarify reasons why you may need a physical therapist to do a balance and gait assessment?

Authors Response: Gait analysis examines an individual's ambulatory pattern and allows for the measurement of the location and orientation of distinct body segments in space, as well as insight into the forces responsible for the observed segmental displacements. (Page 3, line 108-111)

  1. How do you assess gait analysis?

Authors response:  The introduction has been revised to include the gait analysis methods. Page 3, line 93-98)

  1. What is included in a gait assessment?

Authors response:  The introduction has been revised to include the gait analysis methods. Page 3, line 93-105)

  1. How is a normal gait described?

Authors response:  Gait, a mode of locomotion in which the two legs alternately offer both support and propulsion (Page 3, line 80-81)

  1. Which parameters are measured?

Authors response:  The introduction has been revised to include the the various parameters measured during gait analysis. (Page 3, line 99-105)

Reviewer 2 Report

The has relevant content especially for applicability in clinical practice. There are some reflections below that should be considered.

1)   Correct the first line in the introduction section, which start with a reference without context “Pirker and Katzenschlager, 2017; Cruz-Jimenez, 2017).”

2)   In the introduction section the authors stated that “Gait assessment in physical therapy practice is fundamental for assessing musculoskeletal conditions. Many gait assessment tools are applicable in routine clinical practice for musculoskeletal conditions”, however it is not clear what would this assessment bring additional information for these professionals? Is it applied for all types of patients? Only stating “is fundamental for assessing musculoskeletal conditions” is not enough.

3)   The authors should describe better what they considered a “gait assessment”? There are several ways to assess gait with different goals and outcomes according to the area of physiotherapy (eg, gait speed test, 6MWT, visual analysis and soo...)

4)   Despite the focus of the study being on the musculoskeletal conditions, it would be interesting for the authors to discuss the possible qualitative and quantitative methods for gait assessment in different areas of physical therapy and types of patients.

5)   Only 21.6% of physical therapists work with patients with musculoskeletal conditions. Would it be possible for the authors to detail more what “mixed patients” would be? Perhaps this data would help to understand why gait analysis is not being used so much by these physiotherapists, that is, should all types of patients be submitted to gait assessment? Should gait tests performed in all musculoskeletal diseases?

6)   The authors presented the results and were discussing simultaneously which made the reading a little confusing to follow, long and repeated a lot of numbers that were already described in the tables. While the discussion section only contains “Study Restrictions” and “Recommendations and Future Direction”. I suggest better distribution of Results and Discussion sections, maybe the discussion could be divided according to types of gait tests.

Author Response

Response to 2nd Reviewer comments

  1. Correct the first line in the introduction section, which start with a reference without context “Pirker and Katzenschlager, 2017; Cruz-Jimenez, 2017).”

Authors Response: As desired the necessary corrections have been done. 

  1. In the introduction section the authors stated that “Gait assessment in physical therapy practice is fundamental for assessing musculoskeletal conditions. Many gait assessment tools are applicable in routine clinical practice for musculoskeletal conditions”, however it is not clear what would this assessment bring additional information for these professionals? Is it applied for all types of patients? Only stating “is fundamental for assessing musculoskeletal conditions” is not enough.

 Authors Response: The statement has been revised to make it more comprehensible. (page 3 line 95-98)

  1. The authors should describe better what they considered a “gait assessment”? There are several ways to assess gait with different goals and outcomes according to the area of physiotherapy (eg, gait speed test, 6MWT, visual analysis and soo...)

 Authors Response: As desired the details have been added. (page 3 line 107-109)

  1. Despite the focus of the study being on the musculoskeletal conditions, it would be interesting for the authors to discuss the possible qualitative and quantitative methods for gait assessment in different areas of physical therapy and types of patients.

 Authors Response: We did not discriminate between the types of gait analysis since the goal of this research was to determine the view of physical therapists in general on the uses and challenges encountered while conducting gait assessments.

  1. Only 21.6% of physical therapists work with patients with musculoskeletal conditions. Would it be possible for the authors to detail more what “mixed patients” would be? Perhaps this data would help to understand why gait analysis is not being used so much by these physiotherapists, that is, should all types of patients be submitted to gait assessment? Should gait tests performed in all musculoskeletal diseases?

      Authors Response: The purpose of this research was to discover physical therapists' perceptions of the uses and obstacles they experience while doing gait assessments on orthopedic patients. As a result, we inquired if they exclusively handle orthopedic patients or other sorts of patients in need of rehabilitation.

  1. The authors presented the results and were discussing simultaneously which made the reading a little confusing to follow, long and repeated a lot of numbers that were already described in the tables. While the discussion section only contains “Study Restrictions” and “Recommendations and Future Direction”. I suggest better distribution of Results and Discussion sections, maybe the discussion could be divided according to types of gait tests.

Authors Response: The results and discussion is revised to make it more coherent and aligned with the scope of the study.

Reviewer 3 Report

It is a well written and well designed manuscript. The results of the study might give a significant message about using the gait analysis tools in the physical therapy settings in Saudi Arabia for health policy makers, directors, owners and physiotherapists. 

The authors should try to cover at least 80% of the PTs working in these fields so that They can obtain more reliable and valid data.

Author Response

 Response to 3rd Reviewer comments

None required

Round 2

Reviewer 1 Report

The manuscript has been significantly improved.

Author Response

None required

Reviewer 2 Report

The authors increased the theoretical background of the study in the Introduction section only. Most of my concerns were answered, however, Figure 1 is confusing and not self-explanatory, as well as the impact for the scientific community and professionals who do not always need to assess gait is still not very well established.

Author Response

As suggested by the reviewer, the result was further enhanced by removing the figure and providing Table 3, which clearly presents the regression findings. The discussion was further reviewed and streamlined with the scope of the study, and the relevance of utilization of the gait assessment tool was added.